# Effects of Deficit Irrigation and Anti-Stressors on Water Productivity, and Fruit Quality at Harvest and Stored 'Murcott' Mandarin

Hayam M. Elmenofy [1], Harlene M. Hatterman-Valenti [2,*], Islam F. Hassan [3] and Mahmoud Mohamed Abdalla Mahmoud [4,*]

1   Fruit Handling Department, Horticulture Research Institute, Agricultural Research Center, Giza 12619, Egypt; dr.hmoustafa2015@gmail.com
2   Department of Plant Sciences, North Dakota State University, Fargo, ND 58108-6050, USA
3   Water Relations and Field Irrigation Department, Agricultural and Biological Research Institute, National Research Center, Giza 12622, Egypt; if.hassan@nrc.sci.eg
4   Water Requirements and Field Irrigation Research Department, Soils, Water and Environment Research Institute, Agricultural Research Center, Giza 12112, Egypt
*   Correspondence: h.hatterman.valenti@ndsu.edu (H.M.H.-V.); mahmoud_abdalla96@yahoo.com (M.M.A.M.)

**Abstract:** A two-year experiment using 'Murcott' mandarin (*Citrus reticulata*) was conducted under deficit irrigation (DI) strategies with an anti-stressor application and then stored. The three DI regimes were 70% crop evapotranspiration (70% $ET_C$), 85% ETc, and full irrigation (100% ETc). Anti-stressor treatments consisted of a foliar application of either sodium nitroprusside (SNP), selenium nanoparticles (NanoSe), microalgae (*Spirulina platensis*), or a non-treated control. Mean water productivity values were highest for trees at 70% ETc and 85% ETc when they were sprayed with microalgae, which was 42% and 51% greater, respectively, compared to control trees at 100% ETc, but only 6.5% and 13% greater, respectively, compared to trees sprayed with microalgae at 100% ETc. Trees sprayed with microalgae at 100% ETc or 85% ETc had the highest and second-highest fruit yields. The percentage of fruit water loss after 15 d storage at either 1.5 °C + 90% RH or $16 \pm 2$ °C + 60–65% RH was reduced at all DI regimes when trees were sprayed with an anti-stressor compared to control trees at the same DI regime. Trees sprayed with 150 µmol L$^{-1}$ SNP had the only fruit peel that maintained the carotenoid content after cold storage. In general, trees sprayed with SNP were most successful at each DI regime for reduced Malondialdehyde (MDA), but after 15 d of cold storage, only trees sprayed with 150 µmol L$^{-1}$ SNP at 85% ETc or 100% ETc and/or trees sprayed with NanoSe at 70% ETc had fruit that maintained low MDA content. Catalase and peroxidase enzyme activities and proline content were enhanced when trees were sprayed with 150 µmol L$^{-1}$ SNP during DI strategies. This study demonstrates the effectiveness of deficit irrigation combined with anti-stressor applications to improve water productivity, fruit yields, and post-storage quality of 'Murcott' mandarin trees. These findings offer valuable insights into sustainable citrus production under limited water resources.

**Keywords:** deficit irrigation; anti-stressors; fruit quality; storability; mandarin fruit





## 1. Introduction

Citrus is a member of the Rutaceae family and one of the most extensively cultivated tropical and subtropical fruit trees that thrive in semi-arid environments [1]. Globally, approximately 157 million Mg of citrus is produced each year, and about 4.6 million Mg is produced in Egypt, with approximately 2.6 million Mg shipped from Egypt [2]. Mandarin production increased from 797,000 Mg in 2010 to 1.1 million Mg in 2020. Thus, Egypt is self-sufficient in citrus production and exports surplus amounts.

The projected impacts of climate change threaten global food security [3], as well as an increase in extreme water scarcity that restricts agricultural production and expansion,

especially in the Mediterranean region [4,5] as well as Egypt [6]. As a result, researchers worldwide are looking for innovative solutions that permit more efficient utilization of water resources to conserve water, and increase crop yields and water productivity [7]. New irrigation systems must be developed to cope with limited water supplies [1,8]. In addition, deficit irrigation (DI) is a novel, practical, and applicable irrigation strategy to save irrigation water and enhance water productivity. This means that instead of providing crops with irrigation under complete crop water requirements, the crop is exposed to a specific water stress level either during the whole growing period or during a particular growth stage [9].

The deficit irrigation strategy has been discovered to be a beneficial, sustainable production strategy [10], and has widespread use in various Mediterranean countries [11]. Several contributions, in this respect, have documented the implications of DI techniques in terms of enhanced efficiency of water use and fruit quality in various citrus species such as Mandarin [1,4], sweet orange [8,12,13], lime fruit [14], and grapefruit [15,16].

Under stress conditions, it is critical to undertake techniques to mitigate the adverse impacts of water stress on fruit growth [17]. Interestingly, citrus plants can adapt to drought stress by priming factors such as reactive oxygen species (ROS), reactive nitrogen species (RNS), and chemical agents that may initiate intercellular metabolic responses [18]. Plants also synthesize and accumulate suitable solutes called osmolytes or osmoprotectants, not only to lower cell water potential, but also to increase water extraction [19] and redox molecules (ROS and RNS) that affect cellular, physiological, and molecular levels [20]. Some studies have shown that bio-stimulators such as proline [21], selenium [22], and algae extract [23] could increase plant drought resistance. Recent studies have provided compelling evidence that nitrous oxide (NO) induced adaptation toward drought in citrus plants and shed light on essential details of this phenomenon [18]. Functioning as a signaling molecule or as a free radical, NO provided tolerance to various biotic and abiotic stimuli in plants [24]. In addition, exogenous NO supplementation, such as synthetic sodium nitroprusside (SNP), reduced the negative impacts of abiotic stressors and increased resistance [20]. The ability of SNP to operate as a long-lasting NO reservoir has made SNP the most popular NO donor (generator).

Nanotechnology is an expanding field that has applications in agriculture and plant science as highly reactive nano fertilizers that can penetrate the epidermis and are required to reduce the environmental impact of inorganic fertilizers [25]. Nanomaterials' high ratio of surface area to volume facilitates rapid response, enhancing plant development efficiency [26]. In addition, nano fertilizers can increase a plant's tolerance to abiotic stress, which is an enormous benefit [27].

Nano fertilizers such as selenium nanoparticles (NanoSe) were shown to increase vegetative growth, reproductive growth, and flowering, which resulted in improved pomegranate (*Punica granatum*) fruit yield, quality, and shelf life [28]. Cheng et al. [29] also demonstrated that NanoSe increased crop yields and nutrient quality with less financial outlay than conventional Se fertilizers. Additionally, NanoSe enhanced the tricarboxylic acid cycle, which increased plant growth by increasing amino acid and hormone synthesis, and increased photosynthetic pigments to stimulate growth, even in drought conditions [28,29].

There has also been an increasing demand for biological sources like *Spirulina* microalgae (*Spirulina platensis*), a blue–green algae, as more individuals seek out and value natural remedies. To combat malnutrition, *Spirulina* microalgae may be an integral part of nutritional supplements due to its high phytonutrient and antioxidant content [30,31]. Although *Spirulina* microalgae were recently used in agricultural research, products made from microalgae improve nutrient uptake, crop performance, physiological condition, and tolerance to abiotic stress [32].

To our knowledge, no studies have examined the effect of foliar anti-stressors (NO donor SNP, NanoSe, or *Spirulina* microalgae) under DI and full irrigation on a 'Murcott' Mandarin crop. Consequently, the objective of the current study was to determine the effect of graded levels of irrigation and the ameliorative effect of foliar application of three SNP

concentrations, two *Spirulina* microalgae, and Se nanoparticles on 'Murcott' Mandarin fruit yield, fruit physical and biochemical characteristics, and water productivity. In addition, the effect of the foliar anti-stressors under DI and full irrigation were determined for 'Murcott' Mandarin fruit quality after 15 days of cold storage and shelf life.

## 2. Materials and Methods

### 2.1. Experiment Site and Climatic Conditions

Two consecutive experiments in 2021 and 2022 were conducted on a private orchard located at El Reyad, Kafr El Sheikh Governorate region, Egypt. The orchard consisted of 10-year-old 'Murcott' mandarin budded on sour orange rootstock (*Citrus aurantium*). Trees were spaced on 3.5 m centers with 4 m between rows on clay soil. Typical agricultural procedures for the orchard were applied to the trees. Monthly average climatic parameters were collected from the Sakha agro-meteorological station (Latitude: 31°07′ N, Longitude: 30°57′ E), located 7 km from the experimental orchard (Table 1). Before the initiation of each experiment, soil samples were gathered from the site and analyzed. Field capacity, permanent Wilting point, particle-size distribution, bulk density, and total porosity were determined according to Klute [33] (Table 2). Soil pH was determined in a 1:2.5 suspension, and electrical conductivity was determined in soil paste extract, according to Page et al. [34].

**Table 1.** Average climatic parameters (2021–2022) at Sakha agro-meteorological station (31°07′ N Latitude, 30°57′ E Longitude), Kafr El Sheikh Governorate.

| Parameters | | Jan. | Feb. | Mar. | Apr. | May | Jun. | Jul. | Aug. | Sep. | Oct. | Nov. | Dec. |
|---|---|---|---|---|---|---|---|---|---|---|---|---|---|
| Air temperature (°C) | Max. | 18.6 | 20.8 | 23.2 | 27.1 | 31.2 | 32.6 | 33.8 | 34.0 | 33.4 | 29.5 | 25.2 | 20.9 |
| | Min. | 9.9 | 12.2 | 15.9 | 20.1 | 24.3 | 26.6 | 27.2 | 27.3 | 24.9 | 18.0 | 19.6 | 14.0 |
| | Mean | 14.2 | 16.5 | 19.5 | 23.6 | 27.8 | 29.6 | 30.5 | 30.7 | 29.2 | 23.7 | 22.4 | 17.5 |
| Relative humidity (%) | Max. | 86.3 | 86.0 | 83.5 | 80.5 | 73.9 | 78.2 | 83.8 | 78.5 | 84.4 | 78.5 | 83.7 | 87.4 |
| | Min. | 61.2 | 58.3 | 55.5 | 46.1 | 42.3 | 47.7 | 54.2 | 59.8 | 48.9 | 50.9 | 55.0 | 61.4 |
| | Mean | 73.7 | 72.1 | 69.5 | 63.3 | 58.1 | 62.9 | 69.0 | 69.2 | 66.7 | 64.7 | 69.4 | 74.4 |
| Wind speed (km d$^{-1}$) | Mean | 45.8 | 46.3 | 63.8 | 78.8 | 96.4 | 105.8 | 92.3 | 80.0 | 88.2 | 77.8 | 43.4 | 43.1 |
| Pan evaporation (mm d$^{-1}$) | Mean | 2.0 | 2.2 | 3.7 | 5.1 | 6.8 | 8.0 | 7.7 | 7.0 | 5.8 | 4.1 | 2.0 | 1.8 |
| Radiation (Mj m$^{-2}$ d$^{-1}$) | Mean | 12.3 | 15.2 | 19.1 | 23.1 | 25.8 | 28 | 27.3 | 25.8 | 22.1 | 17.8 | 13.8 | 11.3 |
| Rain (mm) | Mean | 34.4 | 14.3 | 18.6 | 2.9 | 0.0 | 0.0 | 0.0 | 0.0 | 0.0 | 13.3 | 6.7 | 26.5 |
| ETo (mm d$^{-1}$) | Mean | 1.5 | 2.0 | 3.0 | 4.3 | 5.5 | 6.2 | 6.1 | 5.8 | 4.9 | 3.4 | 2.2 | 1.6 |

**Table 2.** Mean values of the experimental site's physical and chemical soil properties as an average of the two growing seasons.

| Soil Depth (cm) | Vol. Field Capacity (%) | Vol. Wilting Point (%) | Bulk Density (Mgm$^{-3}$) | Total Porosity (%) | Sand (%) | Silt (%) | Clay (%) | Texture Class | EC$_e$ (dSm$^{-1}$) | pH 1:2.5 |
|---|---|---|---|---|---|---|---|---|---|---|
| 0–15 | 44.81 | 22.67 | 1.22 | 53.96 | 20.13 | 24.75 | 55.12 | Clayey | 2.25 | 7.96 |
| 15–30 | 43.25 | 21.93 | 1.26 | 52.45 | 19.08 | 25.18 | 55.74 | Clayey | 2.71 | 8.25 |
| 30–45 | 41.08 | 20.55 | 1.38 | 47.92 | 17.74 | 25.59 | 56.67 | Clayey | 3.46 | 8.5 |
| 45–60 | 40.5 | 20.78 | 1.47 | 44.53 | 15.63 | 26.44 | 57.93 | Clayey | 3.89 | 8.67 |
| Mean | 42.41 | 21.48 | 1.33 | 49.81 | 18.15 | 25.49 | 56.36 | Clay | 3.08 | |

## 2.2. Experimental Design and Treatments

The experimental design was a randomized complete block design in a split-plot arrangement with three replicates (Figure 1A). Trees were planted in a unique 2 m wide and 0.8 m tall raised bed to avoid water table level effects (Figure 1B). Subsurface drainage was installed before tree planting and checked monthly to make sure the water table was always deeper than 1.5 m. The main plots were the irrigation treatments which consisted of nine homogeneous Mandarin rows that were 73.5 m long. Irrigation treatments were used at every stage of development and consisted of 70% of the crop evapotranspiration (ETc) (70% ETc), 85% of ETc (85% ETc), and full irrigation or 100% of ETc (100% ETc). The drip irrigation system consisted of two lateral lines of 16 mm diam. polyethylene pipes. The laterals were located 60 cm from the tree rows with drippers of 4 Lh$^{-1}$ at 50 cm spacing. Irrigation water was filtered through gravel filters and re-filtered through screen filters. The subplots were the foliar spray treatments which contained three trees. Twice per season (June and July), trees were sprayed using a hand sprayer (Maka Agricultural Machinery Co., Tanta, Egypt) to apply anti-stressor foliar spray treatments of either 50µmol L$^{-1}$ SNP, 100 µmol L$^{-1}$ SNP, 150 µmol L$^{-1}$ SNP, 100 mmol L$^{-1}$ NanoSe, 1% or 2% microalgae (Algae 1%, Algae 2%). All anti-stressors (SNP, microalgae, and NanoSe) were obtained from Microbiology Dept., Sakha Agricultural Research Station, Kafr Elsheikh, Soils, Water, and Environ. Res. Inst., Agric. Res. Center, Giza, Egypt. *Spirulina platensis* was developed in an altered Zarrouk medium [35], and NanoSe were synthesized according to the method by Shalaby [36].

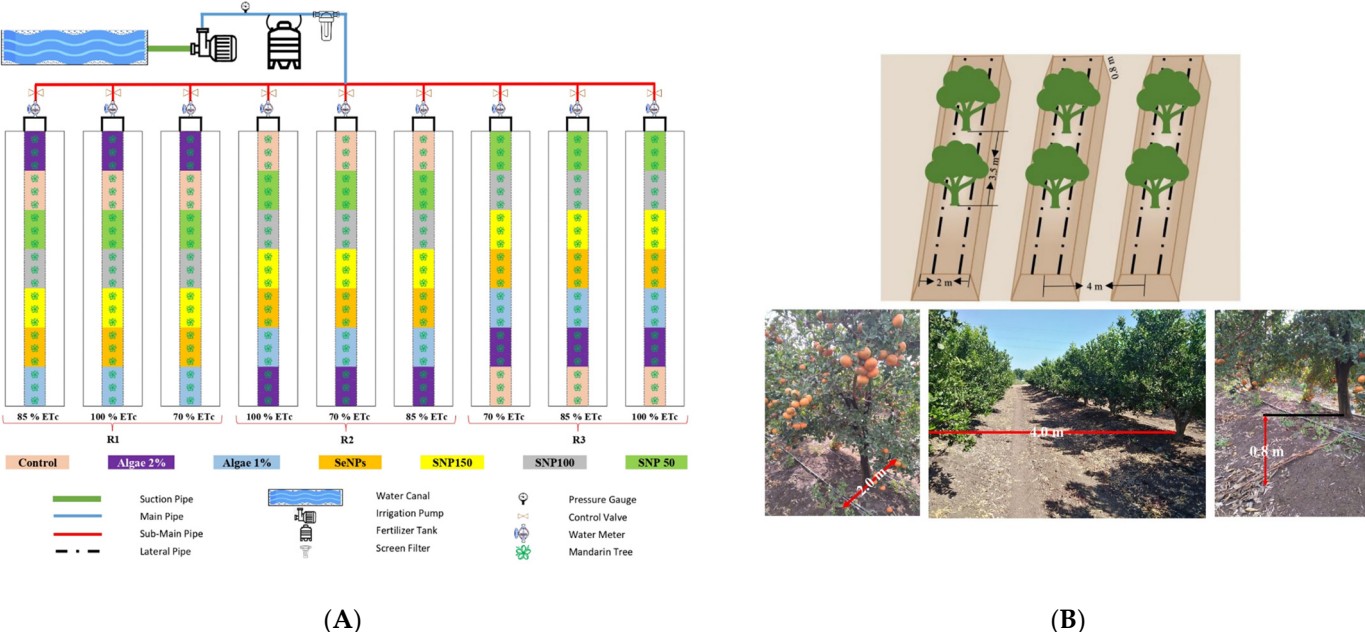

**(A)**                                       **(B)**

**Figure 1.** (**A**). The experiment design and map of the anti-stressor treatments in the 10-year-old 'Murcott' mandarin orchard located at El Reyad, Kafr El Sheikh Governorate region, Egypt. (**B**). The raised-bed design in the 10-year-old 'Murcott' mandarin orchard located at El Reyad, Kafr El Sheikh Governorate region, Egypt.

## 2.3. Data Collection and Measurement

### 2.3.1. Applied Water and Water Productivity

Water for irrigation was piped in from the Nile River, which was close to the test site. The irrigation water's average electrical conductivity was 0.43 dSm$^{-1}$, and the pH was 7.52 across both seasons. The amount of applied water at each irrigation was based on ETc, which was calculated as shown in Equation (1).

$$ETc = ETo \times kc \tag{1}$$

where ETc is the crop evapotranspiration, ETo refers to the reference evapotranspiration, and Kc refers to crop coefficient values obtained from [37]. Reference evapotranspiration was calculated according to the modified Penman–Monteith equation [38] using the mean monthly climatic data from 2016 to 2020 (Table 1) and FAO–CROP WAT 8.0 software [39].

The applied irrigation water (AIW) was calculated as shown in Equation (2), according to Habib [40].

$$AIW = (ETc \times Kr \times Ii + LR) \div Ea \tag{2}$$

where AIW is the applied irrigation water (mm), ETo is reference evapotranspiration ($mmd^{-1}$), Kc is a crop coefficient, Kr is a reduction factor that depends on ground cover, Ii is an irrigation interval, and Ea is an irrigation efficiency (85%). The Ea was estimated from the emitter uniformity coefficient (0.95) multiplied by the drip irrigation efficiency coefficient (0.90), and LR was calculated according to Ayers and Westcot [41], as shown in Equation (3).

$$LR = ECW \div (5ECe - ECw) \tag{3}$$

where LR is the minimum leaching requirement needed to control salts with ordinary surface irrigation methods, ECw is the electrical conductivity of the irrigation water in millimhos per centimeter at 25 °C, and ECe is the electrical conductivity of the saturation paste extract of the soil reported in millimhos per centimeter at 25 °C.

Applied water was determined as documented by Giriappa [42] in Equation (4).

$$AW = AIW + ER \tag{4}$$

where AW is applied water, AIW is applied irrigation water, and ER is the effective rainfall = incident rainfall × 0.7 [43].

Water productivity (WP) was calculated according to Rajanna et al. [44], as shown in Equation (5).

$$WP = MY \div AIW \tag{5}$$

where WP is the water productivity ($kg\ m^{-3}$), MY is the marketable yield ($kg\ ha^{-1}$), and AIW is the irrigation water applied ($m^3\ ha^{-1}$).

### 2.3.2. Fruit Yield and Quality Components

The last week of December marked harvest time in both years. To calculate the total fruit yield ($Mg\ ha^{-1}$), the average fruit weight (g) was multiplied by the number of fruits produced by each tree and then multiplied by the number of trees per hectare. The fruit's volume (mL) was determined using the water displacement method and fruit peel thickness (mm). Random samples of three uniform fruits per tree (nine fruits/replicate) were selected to analyze harvest data. Additional samples of 10 undamaged and decay-free fruits per replicate (30 fruits/treatment) were collected, washed with tap water containing $1\ mgL^{-1}$ chlorine, and allowed to air-dry at room temperature (22–24 °C) for 30 min. Samples were divided into two subsamples, each placed in a cardboard box and kept at 1.5 °C and 90% RH to assess fruit qualities at harvest and again after 15 days of storage, or kept at room condition (16 ± 2 °C and 80–85% RH) for 15 days to assess shelf-life period. Fruit juice weight (g/fruit) was measured after homogenizing the fruit pulp in a blender and filtering it. The juice % was determined using the formula: (juice weight) ÷ (juice and pulp weight) 100%. Approximately 50 g of pulp fruit was extracted to determine soluble solids content (SSC%) using a digital refractometer (RFM 340–T, KEM Kyoto Electronics Manufacturing Co., Ltd., Tokyo, Japan) at 25 °C, and to determine total acidity (TA) as citric acid (%) using an automated titration device (TitroLine, TL 5000, SI Analytics, Weinheim, Germany) according to the Association of Official Agriculture Chemists (AOAC) [45].

After 15 days of cold storage, cooled fruit subsamples were weighed on a digital balance (EK600H—A&D Company Ltd., Tokyo, Japan) to determine weight loss percentage based on the difference between the fruits' preliminary and final weights. Fruit weight loss percentage was also determined for fruit subsamples stored at room temperature.

Using the following formula from Jannatizadeh [46], the degree of chilling injury was calculated based on the 4-point hedonic scale developed by Sayyari et al. [47] for husk browning of the fruit surface.

$$CI\ (\%) = \sum [(\text{value of hedonic scale}) \times (\text{fruit number at CI scale})]/(4 \times \text{total number of fruit}) \times 100.$$

Thiobarbituric acid-reactive substances (TBARS) were used to approximate the lipid peroxidation in the cellular membrane by measuring the Malondialdehyde content (MDA) ($\mu$mole g$^{-1}$ FW) in fruit peel [48,49].

To determine the carotenoid content "$\mu$g mL$^{-1}$", 1 g of fruit peel was dissolved in 10 mL of 80% acetone and left at room temperature in a dark bottle for 24 h until measured by a spectrophotometer (UV/visible spectrophotometer Libra SS0PC, Thermo Fisher Scientific, Waltham, MA, USA). The absorbances were recorded at 663, 646, and 470 nm for chlorophyll a, chlorophyll b, and carotenoids, respectively, and total contents were calculated according to the following Equations (6)–(8):

$$\text{Chlorophyll (a)} = 12.21\ A663 - 2.81\ A646. \tag{6}$$

$$\text{Chlorophyll (b)} = 20.13\ A646 - 5.03\ A663. \tag{7}$$

$$\text{Total carotenoids} = ((1000\ A470) - (3.27 \times \text{chlorophyll a} + 104 \times \text{chlorophyll b}))/198. \tag{8}$$

where A is the optical density at the specified wavelength [50].

To determine the proline content "$\mu$mole g$^{-1}$ fresh weight (FW)", a 0.5 g sample of fresh fruit peel was homogenized in 10 mL of 3% sulfosalicylic acid, then centrifuged at 11,500$\times$ *g* for 15 min. Two milliliters of filtrate were mixed with 2 mL acid ninhydrin and 2 mL glacial acetic acid; it was then incubated for 1 h at 100 °C, cooled in an ice bath, and 4 mL toluene was added. The mixture was cooled in an ice bath, then 4 mL of toluene was added while stirring for 60 s. Absorption of chromophores was evaluated using the same spectrophotometer emitting radiation at 520 nm. The proline level was calculated using the method of Bates et al. [51], where the standard curve and the specific content were calculated on a fresh weight basis as follows:

$$\mu\text{mole proline g}^{-1}\text{ fresh weight} = [\ (\mu\text{g proline mL}^{-1} \times \text{mL toluene}/115.5\ \mu\text{g}/\mu\text{mole})] \div (\text{g sample}/5).$$

### 2.3.3. Antioxidant Enzyme Activity

To determine the activity of catalase enzyme (CAT) "$\mu$mole g$^{-1}$ FW min$^{-1}$", the method described by Aebi [52] was used. The reaction solution (3 mL) contained 1.5 mL of 100 mM potassium phosphate buffer (pH 7), 0.5 mL of 75 mM $H_2O_2$, 0.05 mL enzyme extraction, and distilled water to make up the volume to 3 mL. The reaction started by adding $H_2O_2$, and a decrease in absorbance was recorded at 240 nm ($\varepsilon = 36$ mM$^{-1}$·cm$^{-1}$) for 1 min. Enzyme activity was computed by calculating the amount of $H_2O_2$ decomposed.

The activity of peroxidase enzyme (POD) "$\mu$mole g$^{-1}$ FW min$^{-1}$" was measured by following the change of absorption at 470 nm due to guaiacol oxidation. Peroxidase activity was measured according to Polle et al. [53]. The reaction mixture was prepared by adding 100 mM potassium phosphate buffer (pH 7), 10 mM $H_2O_2$, 20 mM guaiacol, 0.05 mL enzyme extraction, and distilled water to make up the volume to 3 mL. The reaction started by adding $H_2O_2$, and a decrease in absorbance was recorded at 470 nm ($\varepsilon = 26.6$ mM$^{-1}$·cm$^{-1}$) for 1 min. Enzyme activity was computed by calculating the amount of $H_2O_2$ decomposed.

### 2.4. Statistical Analysis

The PROC Univariate in SAS ver. 9.4 (SAS Institute, Cary, NC, USA) was used to test if all data met assumptions of normality before the analysis of variance was performed. The generalized linear mixed model (PROC GLIMMIX) in SAS was performed with a specific distribution of data set for the analysis of variance. Years were combined since their error mean squares differed by less than a factor of 10 and were considered homogenous. When

the F-test for treatments was significant ($\alpha < 0.05$), means were compared by using multiple *t*-tests using the lines options statement in PROC GLIMMIX. The generalized linear mixed model was used for all data sets for the analysis of variance since treatment and year were considered fixed effects and random effects, respectively.

## 3. Results

### 3.1. Applied Water and Water Productivity of 'Murcott' Mandarin Fruit

Applied water increased with increased plant growth activity and air temperature due to increased crop evapotranspiration (Table 3). The highest values of applied water were obtained during the flowering to fruit growth stage, while the lowest values were obtained during winter. Analysis of the total amount of applied water indicated that the highest amount of applied water occurred with 100% ETc and was significantly reduced by 15% and 30% at 85% ETc and 70% ETc, respectively, compared to 100% ETc.

**Table 3.** Monthly and seasonal applied irrigation water as affected by anti-stressor treatments under DI of Murcott fruit.

| | **Monthly Applied Water (mm)** | | | | | | | | | | | | **Seasonal** | |
|---|---|---|---|---|---|---|---|---|---|---|---|---|---|---|
| **Irrigation** | **Jan.** | **Feb.** | **Mar.** | **Apr.** | **May** | **Jun.** | **Jul.** | **Aug.** | **Sep.** | **Oct.** | **Nov.** | **Dec.** | **(mm)** | **(m³ha⁻¹)** [Z] |
| 100% ETc | 29 | 35 | 70 | 95 | 119 | 118 | 110 | 98 | 81 | 59 | 38 | 30 | 882 | 8834 a |
| 85% ETc | 25 | 29 | 60 | 81 | 101 | 101 | 93 | 84 | 69 | 50 | 32 | 26 | 750 | 7500 b |
| 70% ETc | 20 | 24 | 49 | 66 | 83 | 83 | 77 | 69 | 57 | 41 | 27 | 21 | 618 | 6177 c |

[Z] Means within the column with the same letter do not statistically differ according to Duncan's multiple range test at $p \leq 0.05$.

Water productivity was influenced by irrigation treatments and anti-stressor treatments (Figure 2). The most effective increase in water productivity values was obtained when Algae 2% was applied to trees under 85%ETc, followed by Algae 2% applied to trees under 70% ETc. In contrast, the lowest water productivity value was obtained at 100% ETc without anti-stressor foliar spray on trees.

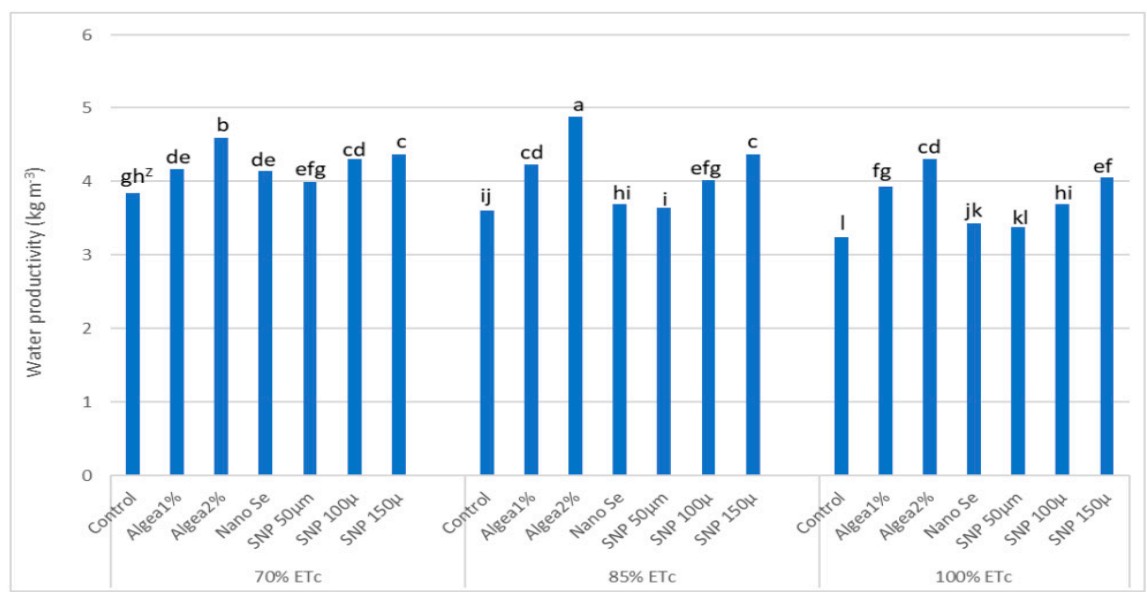

**Figure 2.** Effects of deficit irrigation and foliar anti-stressor treatments on water productivity of 'Murcott' fruit at harvest combined twoyears at El Reyad, Kafr El Sheikh Governorate region, Egypt.
[Z] Means followed by the same letter are similar according to multiple *t*-tests at $\alpha = 0.05$ level of significance.

### 3.2. Fruit Yield and Quality Components

Trees that did not receive an anti-stressor spray showed that yield decreased as DI regimes increased from full irrigation or 100% ETc to 70% ETc (Table 4). The highest fruit yield was obtained when trees were sprayed with Algae 2% under 100% ETc, followed by trees sprayed with Algae 2% under the 85% ETc regime. The lowest fruit yield occurred at 70% ETc when trees were not sprayed with an anti-stressor.

**Table 4.** Effects of deficit irrigation and foliar anti-stressor treatments on 'Murcott' fruit volume and yield at harvest combined two years at El Reyad, Kafr El Sheikh Governorate region, Egypt.

| Treatments | Fruit Volume (mL) | | | Fruit Yield (Mgha$^{-1}$) | | |
|---|---|---|---|---|---|---|
| | 70% ETc | 85% ETc | 100% ETc | 70% ETc | 85% ETc | 100% ETc |
| Control | 97.5 k [Z] | 108.8 j | 139.1 fg | 2.37 l [Z] | 2.70 hi | 2.85 fg |
| Algae 1% | 136.0 g | 144.3 ef | 159.9 db | 2.57 ijk | 3.17 d | 3.46 c |
| Algae 2% | 152.6 c | 167.1 a | 169.5 a | 2.83 g | 3.65 b | 3.80 a |
| Nano Se | 96.5 k | 139.0 fg | 149.8 cde | 2.55 jk | 2.77 gh | 3.03 e |
| SNP 50 μm | 99.4 k | 116.0 hi | 144.8 def | 2.46 kl | 2.72 gh | 2.98 ef |
| SNP 100 μm | 101.2 k | 122.0 h | 145.6 def | 2.65 hij | 3.01 1e | 3.26 d |
| SNP 150 μm | 111.7 ij | 146.8 cde | 151.0 cd | 2.69 hi | 3.27 d | 3.57 bc |

[Z] Means followed by the same letter for each parameter measured are similar according to multiple *t*-tests at $\alpha = 0.05$ level of significance.

'Murcott' fruit volumes and fruit peel thicknesses were affected by DI regimes and anti-stressor treatments (Table 4, Figure 3), while fruit juice percentage was only affected by anti-stressor treatments (Table 5). Like fruit yield, fruit volume decreased as DI regimes increased from full irrigation or 100% ETc to 70% ETc when trees were not sprayed with an anti-stressor spray (Table 4). The highest fruit volumes were obtained when trees were sprayed with Algae 2% at 100% ETc or 85% ETc, while the lowest fruit volumes occurred when trees were sprayed with NanoSe, 50 μmol L$^{-1}$ SNP, 100 μmol L$^{-1}$ SNP, or not sprayed with an anti-stressor (control) at 70% ETc.

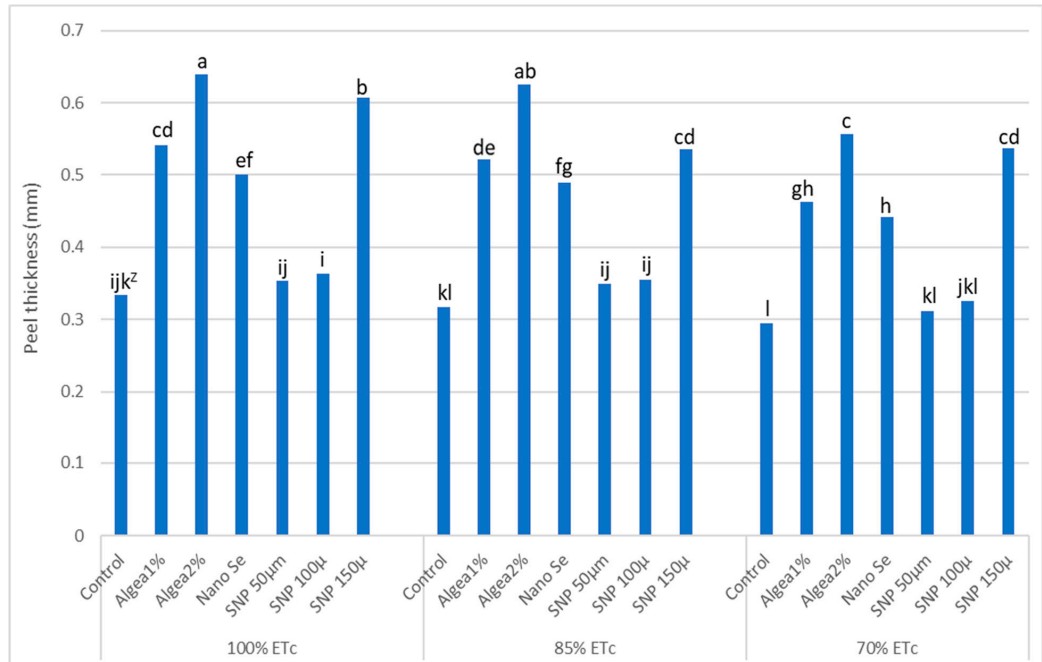

**Figure 3.** Effects of deficit irrigation and foliar anti-stressor treatments on 'Murcott' fruit peel thickness at harvest combined twoyears at El Reyad, Kafr El Sheikh Governorate region, Egypt. [Z] Means followed by the same letter are similar according to multiple *t*-tests at $\alpha = 0.05$ level of significance.

Table 5. Effect of anti-stressor treatments on juice percentage at harvest when combined over deficit irrigations and years at El Reyad, Kafr El Sheikh Governorate region, Egypt.

| Treatments | Juice | |
|---|---|---|
| | % | |
| Control | 56.5 | c [z] |
| Algea 1% | 65.4 | a |
| Algea 2% | 67.0 | a |
| Nano Se | 58.0 | bc |
| SNP 50 μm | 57.0 | c |
| SNP 100 μm | 57.1 | c |
| SNP 150 μm | 60.5 | b |

[z] Means in the same column followed by the same letter are similar according to multiple *t*-tests at α = 0.05 level of significance.

The fruit peel was thickest when trees were sprayed with Algae 2% at either 70% ETc or 85%ETc (Figure 3). The thinnest fruit peel occurred when trees were not sprayed with an anti-stressor and the 100% ETc DI regime. However, this low peel thickness did not differ from fruit peel thickness when trees were not sprayed with an anti-stressor at 85% ETc, or when trees were sprayed with either 50 μmol L$^{-1}$ SNP or 100 μmol L$^{-1}$ SNP at 100% ETc.

Trees sprayed with either Algae 1% or Algae 2% had fruit with the highest juice percentage, while the control trees had the lowest juice percentage (Table 5). However, the lowest juice percentage with the control trees was not different from the fruit juice percentage when trees were sprayed with either NanoSe, 50 μmol L$^{-1}$ SNP, or 100 μmol L$^{-1}$ SNP.

'Murcott' fruit SSC% at harvest was affected by DI regimes and anti-stressor treatments (Figure 4). The highest fruit SSC% at harvest occurred with control trees at 70% ETc, which was greater than the fruit SSC% for trees receiving any other anti-stressor treatment and DI regime combination. Trees under 70% ETc and sprayed with Algae 2% had the next highest fruit SSC% but did not differ from fruit SSC% when trees were under either 85% ETc or 100% ETc and received no anti-stressor spray. Fruit SSC% was lowest when trees were sprayed with SNP at 150 μmol L$^{-1}$ regardless of the DI regime. Fruit SSC% after 15 days of storage did not differ from the data at harvest, indicating that anti-stressor treatments had no influence on fruit SSC% after the fruit was harvested and stored.

'Murcott' fruit acidity varied in response to DI regimes and anti-stressor treatments (Table 6). The highest fruit acidity occurred when trees were sprayed with SNP at 150 μmol L$^{-1}$ under either 85% ETc or 100% ETc. In contrast, the lowest fruit acidity occurred when trees were under 70% ETc and sprayed with either Algae 1% or not sprayed with an anti-stressor. However, this low fruit acidity did not differ from control trees at 85% ETc or 100% ETc. The change in 'Murcott' fruit acidity after 15 days of storage varied in response to DI regimes and anti-stressor treatments. The greatest decrease in fruit acidity occurred when control trees were under 70% ETc. However, this fruit acidity decrease after storage did not differ from fruit acidity changes after storage when trees were sprayed with SNP at 100 μmol L$^{-1}$ or SNP at 150 μmol L$^{-1}$ under 85% ETc, as well as trees sprayed with Algae 1%, SNP at 50 μmol L$^{-1}$, or SNP at 150 μmol L$^{-1}$ under 100%ETc.

A post-harvest water loss has been shown to drastically impact fruit durability and marketability [54]. 'Murcott' fruit water loss after 15 days of cold storage (stored at 1.5 °C and 90% RH) and after 15 days of shelf-life (stored at 16 ± 2 °C and 60–65% RH) varied in response to DI regimes and anti-stressor treatments (Figures 5 and 6). Fruit from control trees under 70% ETc had the most water loss (%weight loss) after 15 days of cold storage or 15 days of shelf-life, which was greater than the fruit water loss from trees receiving any other DI regime and foliar anti-stressor treatment combination. In contrast, the lowest fruit water loss after 15 days of cold storage or 15 days of shelf-life occurred when trees were sprayed with Algae 2% at 100% ETc. As the DI increased from 100% ETc to 70% ETc, trees that were sprayed with Algae 2% produced fruit with the lowest fruit water loss after 15 days of cold storage or 15 days of shelf–life, which was less than the fruit water loss from control trees at the same DI regime. However, the low fruit water loss after 15 days of

cold storage for trees at 85% ETc that were sprayed with Algae 2% did not differ from the low fruit water loss after 15 days of cold storage for trees that were sprayed with 150 µmol $L^{-1}$ SNP. In addition, the low fruit water loss after 15 days of cold storage or 15 days of shelf-life for trees at 70% ETc that were sprayed with Algae 2% did not differ from the low fruit water loss after 15 days of cold storage or 15 days of shelf-life for trees at 70% ETc that were sprayed with 150 µmol $L^{-1}$ SNP.

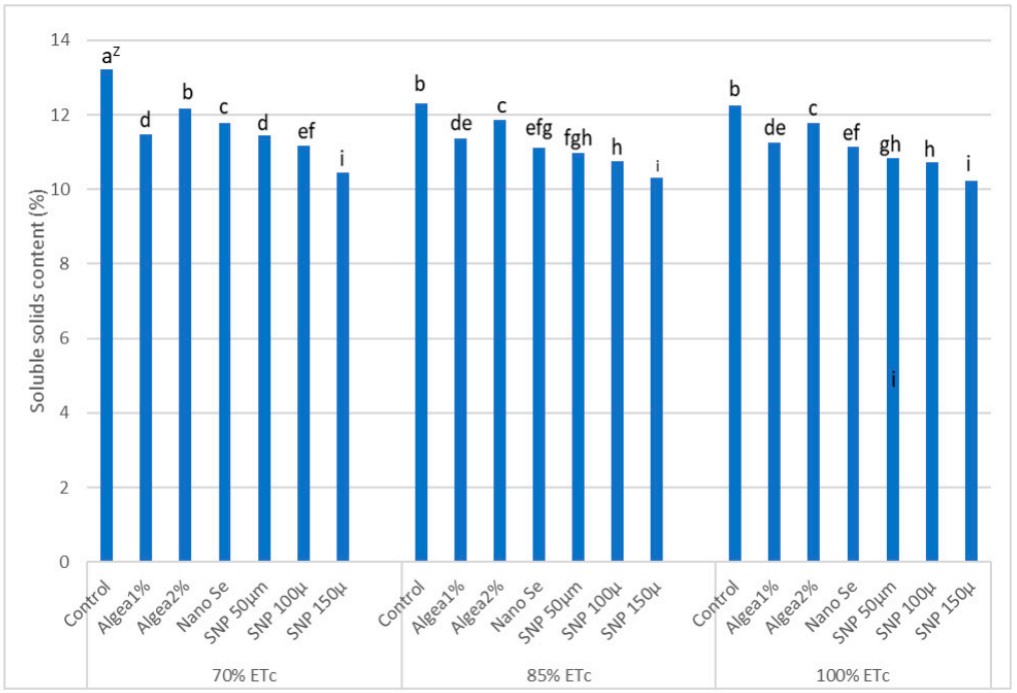

**Figure 4.** Effects of foliar anti-stressor treatments and deficit irrigations (70% ETc; 85% ETc; 100% ETc) on fruit SSC% at harvest combined twoyears at El Reyad, Kafr El Sheikh Governorate region, Egypt. [Z] Means followed by the same letter are similar according to multiple *t*-tests at $\alpha$ = 0.05 level of significance.

**Table 6.** Effects of deficit irrigation and foliar anti-stressor treatments on 'Murcott' fruit acid percent at harvest and change% after 15 d of storage at 1.5 °C and 90% RH combined two years at El Reyad, Kafr El Sheikh Governorate region, Egypt.

| Treatments | Total Acidity (%) | | | Total Acidity Change (%) | | |
|---|---|---|---|---|---|---|
| | 70% ETc | 85% ETc | 100% ETc | 70% ETc | 85% ETc | 100% ETc |
| Control | 1.21 k [Z] | 1.39 ij | 1.39 ij | −49.1 h [Z] | −29.0 abcdef | −28.0 abc |
| Algae 1% | 1.32 jk | 1.48 hi | 1.73 def | −23.5 ab | −29.0 abcdef | −39.0 defgh |
| Algae 2% | 1.49 hi | 1.62 fgh | 1.68 ef | −34.0 bcdef | −35.0 cdefg | −32.0 abcdef |
| Nano Se | 1.48 hi | 1.66 efg | 1.81 de | −37.0 cdefg | −32.0 abcdef | −36.0 cdefg |
| SNP 50 µm | 1.52 ghi | 1.60 fgh | 1.85 cd | −23.0 a | −28.0 abcd | −39.0 efgh |
| SNP 100 µm | 1.78 de | 1.99 bc | 2.05 b | −29.0 abcde | −40.0 fgh | −37.0 cdefg |
| SNP 150 µm | 2.01 b | 2.53 a | 2.53 a | −33.0 abcdef | −45.0 gh | −45.0 gh |

[Z] Means followed by the same letter for each parameter measured are similar according to multiple *t*-tests at $\alpha$ = 0.05 level of significance.

In the current study, 'Murcott' fruit peel proline content varied in response to DI regimes and anti-stressor treatments (Figure 7). At harvest, the proline level generally increased as the DI increased from 100% ETc to 70% ETc, except for the control trees at 85% ETc and 70% ETc, which did not differ. The highest proline content occurred when trees at 70% ETc were sprayed with 100 µmol $L^{-1}$ SNP or 150 µmol $L^{-1}$ SNP, while the lowest proline level occurred with control trees at 100% ETc. However, this low fruit peel proline level did not differ from those for trees at 100% ETc and sprayed with either Algae 1% or

NanoSe. Interestingly, the percent change in proline content between harvest and after 15 days of cold storage was not significant for DI regimes, foliar anti-stressor treatments, or the interaction of DI regime and anti-stressor treatment.

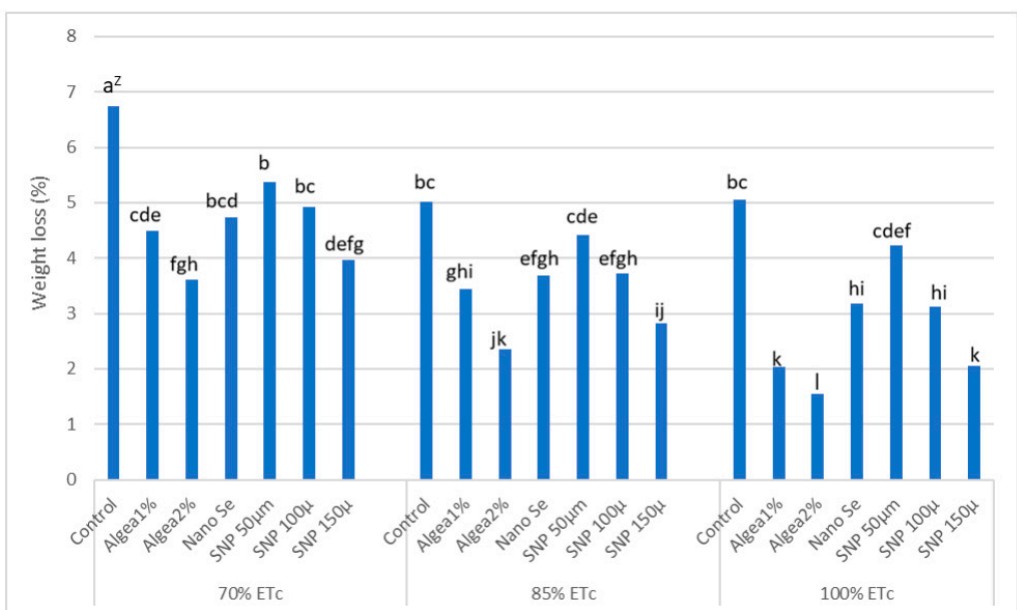

**Figure 5.** Percent fruit water loss after 15 d of storage at 1.5 °C and 90% RH, as affected by anti-stressor foliar treatments and deficit irrigation regimes, and combined twoyears at El Reyad, Kafr El Sheikh Governorate region, Egypt. [Z] Means followed by the same letter are similar according to multiple *t*-tests at $\alpha$ = 0.05 level of significance.

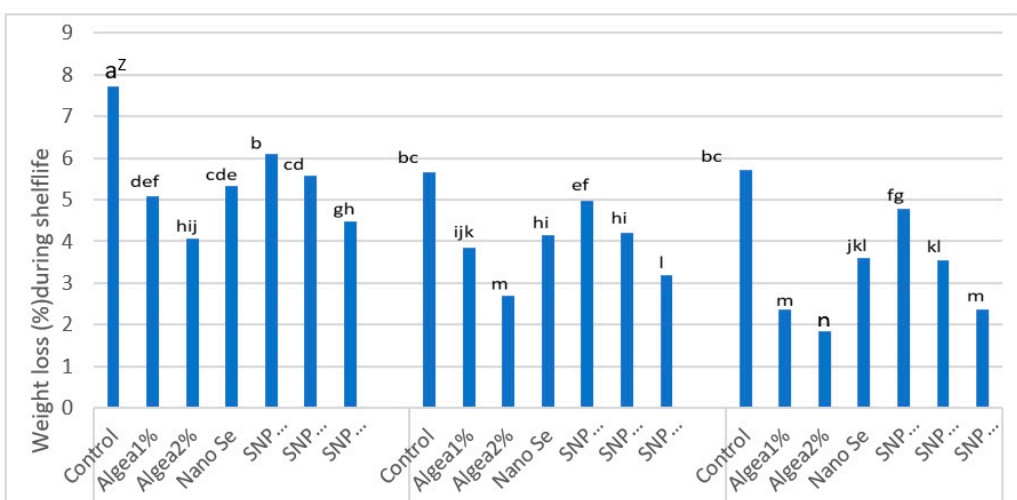

**Figure 6.** Percent fruit water loss after 15 d of shelf-life (stored at 16 ± 2 °C and 60–65% RH, as affected by anti-stressor foliar treatments and deficit irrigation regimes, and combined two years at El Reyad, Kafr El Sheikh Governorate region, Egypt. [Z] Means followed by the same letter are similar according to multiple *t*-tests at $\alpha$ = 0.05 level of significance.

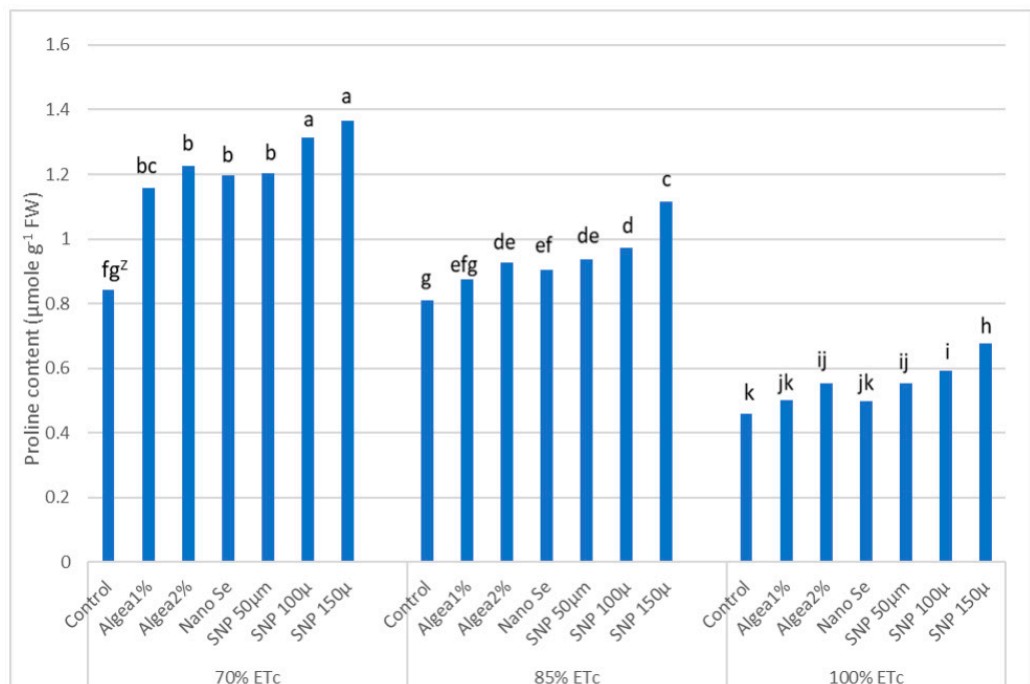

**Figure 7.** Effects of deficit irrigation and anti-stressor treatments on 'Murcott' fruit peel proline content "µmole g–1FW" at harvest, and combined two years at El Reyad, Kafr El Sheikh Governorate region, Egypt. [Z] Means followed by the same letter are similar according to multiple *t*-tests at $\alpha$ = 0.05 level of significance.

'Murcott' fruit peel carotenoid content at harvest and the percent change after 15 days of cold storage was only influenced by foliar anti-stressor treatments (Table 7). The highest carotenoid content occurred when trees were sprayed with 150 µmol L$^{-1}$ SNP, which was significantly greater than the carotenoid content from trees sprayed with any other anti-stressor, as well as control trees. The percent change in carotenoid contents between harvest and after 15 days of cold storage increased only when trees were sprayed with 150 µmol L$^{-1}$ SNP, which was significantly greater than the percent carotenoid content change from trees sprayed with any other anti-stressor, as well as control trees.

**Table 7.** Effect of anti-stressor treatments on carotenoid content "µg mL$^{-1}$" at harvest; % change in carotenoid content, and peroxidase enzyme (POD) activity after 15 d storage at 1.5 °C and 90% RH, when combined over deficit irrigations and years at El Reyad, Kafr El Sheikh Governorate region, Egypt.

| Treatments | Carotenoid Content | | Carotenoid Change | | POD Change | |
|---|---|---|---|---|---|---|
| | "µg mL$^{-1}$" | | % | | | |
| Control | 1.91 | a [z] | −6.6 | b | 0.4 | b |
| Algea 1% | 1.34 | bc | −9.0 | b | 5.3 | a |
| Algea 2% | 1.44 | b | −5.0 | b | 5.1 | a |
| Nano Se | 1.40 | bc | −4.1 | b | 5.3 | a |
| SNP 50 µm | 1.30 | bc | −9.8 | b | 5.5 | a |
| SNP 100 µm | 1.22 | c | −10.4 | b | 5.4 | a |
| SNP 150 µm | 0.89 | d | 6.2 | a | 5.9 | a |

[Z] Means in the same column followed by the same letter are similar according to multiple *t*-tests at $\alpha$ = 0.05 level of significance.

'Murcott' fruit peel MDA content at harvest and the percent change after 15 days of cold storage was affected by DI regimes and foliar anti-stressor treatments (Table 8). The highest fruit peel MDA content at harvest and indicator of oxidative damage occurred at 70% ETc when trees were not sprayed with a foliar anti-stressor. In contrast, the lowest fruit

peel MDA content at harvest was 100% ETc when trees were sprayed with 150 µmol L$^{-1}$ SNP. However, this low MDA content did not differ from the MDA content for trees at 100% ETc and sprayed with 100 µmol L$^{-1}$ SNP or for trees at 85% ETc and sprayed with 150 µmol L$^{-1}$ SNP.

**Table 8.** Effects of deficit irrigation and foliar anti-stressor treatments on 'Murcott' peel Malondialdehyde (MDA) content "µmole g$^{-1}$ FW" at harvest, and change % after 15 d of storage at 1.5 °C and 90% RH, combined two years at El Reyad, Kafr El Sheikh Governorate region, Egypt.

| Treatments | MDA Content (µmole g$^{-1}$ FW) | | | MDA Change % | | |
| --- | --- | --- | --- | --- | --- | --- |
| | 70% ETc | 85% Etc | 100% Etc | 70% Etc | 85% Etc | 100% Etc |
| Control | 30.25 a [Z] | 21.79 c | 21.43 c | 43.9 efg [Z] | 77.5 bcd | 61.2 de |
| Algae 1% | 18.95 d | 16.54 defg | 15.13 fgh | 74.8 bcd | 90.1 bc | 75.7 bcd |
| Algae 2% | 25.26 b | 17.99 de | 18.03 de | 43.5 efg | 92.8 b | 78.5 bcd |
| Nano Se | 23.38 bc | 17.03 def | 13.82 h | 26.1 fg | 64.0 cde | 88.4 bcd |
| SNP 50 µm | 15.78 efgh | 15.3 fgh | 10.59 i | 86.3 bcd | 77.0 bcd | 123.9 a |
| SNP100 µm | 15.02 fgh | 10.36 i | 8.54 ij | 77.2 bcd | 124.8 a | 132.9 a |
| SNP150 µm | 14.35 gh | 9.36 ij | 7.52 j | 45.6 ef | 24.9 fg | 16.5 g |

[Z] Means followed by the same letter for each parameter are similar according to multiple *t*-tests at α = 0.05 level of significance.

The greatest percent change in fruit peel MDA content between harvest and after 15 days of cold storage occurred for trees at 100% ETc and sprayed with 50 µmol L$^{-1}$ SNP or 100 µmol L$^{-1}$ SNP, and for trees at 85% ETc and sprayed with 100 µmol L$^{-1}$ SNP (Table 8). In contrast, the lowest percent change in fruit peel MDA content between harvest and after 15 days of cold storage occurred with fruit from trees at 100% ETc and sprayed with 150 µmol L$^{-1}$ SNP, with fruit from trees at 85% ETc and sprayed with 150 µmol L$^{-1}$ SNP, and with fruit from trees at 70% ETc and sprayed with NanoSe.

*3.3. Antioxidant Enzyme Activity*

Upon experiencing DI, the fruit's enzymes (CAT and POD) are responsible for antioxidant protection and are considerably enhanced when exposed to water deficiency conditions. 'Murcott' fruit peel CAT and POD activity at harvest was affected by DI regimes and anti-stressor treatments (Table 9, Figure 8). The highest fruit peel CAT activity occurred with fruit from trees at 70% Etc and sprayed with 150 µmol L$^{-1}$ SNP, which was significantly greater than the fruit peel CAT activity from any other DI regime and anti-stressor combination (Table 8). Similarly, the highest fruit peel POD activity occurred with fruit from trees at 70% Etc and sprayed with 150 µmol L$^{-1}$ SNP (Figure 8). However, this high POD activity did not differ from the fruit POD activity with fruit from trees at 70% Etc and sprayed with either 50 µmol L$^{-1}$ SNP, 100 µmol L$^{-1}$ SNP, or Algae 2%. The lowest fruit POD activity occurred with fruit from control trees at 100% Etc, as well as trees at 100% Etc and sprayed with either Algae 1% or NanoSe.

The percent change in 'Murcott' fruit CAT activity between harvest and after 15 d of cold storage was affected by DI regimes and foliar anti-stressor treatments (Table 8). The fruit CAT activity increased after cold storage for all DI regimes and anti-stressor treatment combinations. The highest percent increase in CAT activity occurred for fruit from trees at 70% Etc and sprayed with Algae 1%. However, this percent increase in CAT activity did not differ for fruit from control trees at 70% Etc, or fruit from trees at 85% Etc and sprayed with either Algae 1%, NanoSe, 50 µmol L$^{-1}$ SNP, or 100 µmol L$^{-1}$ SNP.

The percent change in 'Murcott' fruit POD activity between harvest and after 15 d of cold storage was only affected by foliar anti-stressor treatments (Table 7). Fruit POD activity increased after cold storage for fruit from trees sprayed with an anti-stressor when compared to control trees.

**Table 9.** Effects of deficit irrigation and anti-stressor treatments on 'Murcott' fruit peel catalase enzyme (CAT) activity (µmole g$^{-1}$ FW min$^{-1}$) at harvest, % change after 15 d of storage at 1.5 °C and 90% RH, combined two years at El Reyad, Kafr El Sheikh Governorate region, Egypt.

| Treatments | CAT "µmole g$^{-1}$ FW min$^{-1}$" | | | CAT Change (%) | | |
|---|---|---|---|---|---|---|
| | 70% ETc | 85% ETc | 100% ETc | 70% ETc | 85% ETc | 100% ETc |
| Control | 12.89 f [Z] | 10.92 jk | 10.62 k | 2.88 ab | 1.68 cdef | 0.69 f |
| Algae 1% | 13.01 ef | 11.71 i | 10.86 k | 2.97 a | 2.59 abc | 0.69 f |
| Algae 2% | 14.02 c | 12.73 fg | 11.37 ij | 0.94 ef | 1.26 def | 0.69 f |
| Nano Se | 13.45 de | 12.26 gh | 10.98 jk | 1.77 cde | 2.05 abcd | 0.69 f |
| SNP 50 µm | 14.23 bc | 12.65 fg | 11.39 ij | 1.32 def | 2.36 abc | 1.29 def |
| SNP 100 µm | 14.63 b | 13.49 d | 11.48 i | 1.32 def | 2.01 abcd | 1.29 def |
| SNP 150 µm | 15.16 a | 14.55 b | 11.79 hi | 1.42 def | 1.89 bcd | 1.29 def |

[Z] Means followed by the same letter for each parameter are similar according to multiple *t*-tests at α = 0.05 level of significance.

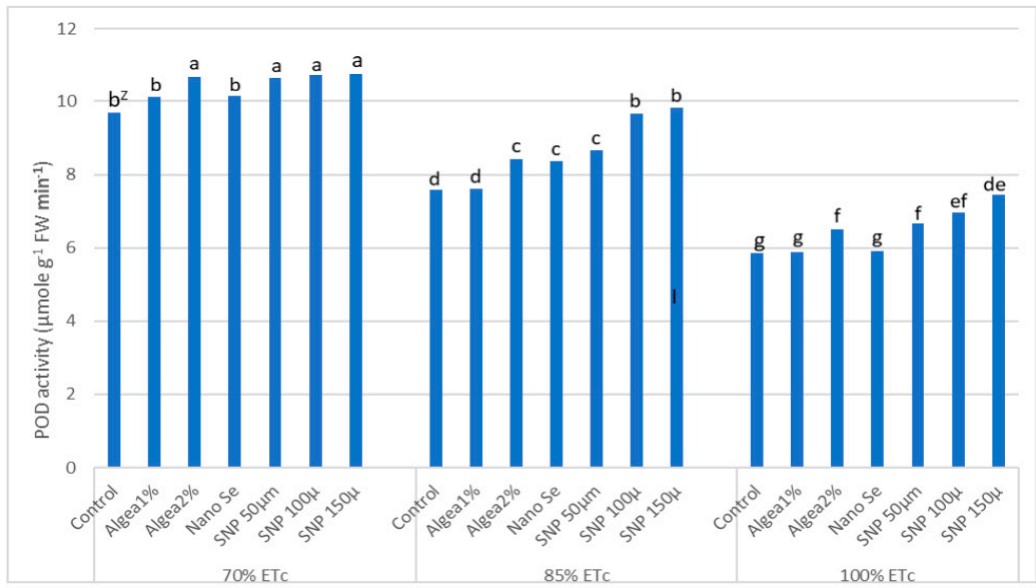

**Figure 8.** Effects of deficit irrigation and foliar anti-stressor treatments on 'Murcott' fruit peel peroxidase enzyme (POD) activity "µmole g$^{-1}$ FW min$^{-1}$" at harvest combined twoyears at El Reyad, Kafr El Sheikh governorate region, Egypt. [Z] Means followed by the same letter are similar according to multiple *t*-tests at α = 0.05 level of significance.

## 4. Discussion

### 4.1. Applied Water and Water Productivity of 'Murcott' Mandarin Fruit

Water deficiency is a severe environmental stress that limits agricultural growth and output, particularly in arid and semi-arid countries. It is crucial to implement strategies to lessen the negative effects of water stress on fruit growth, which can have various physiological effects [17]. In our study, water savings were largely increased by 15% and 30% of 85% ETc and 70% ETc, respectively, compared to 100% ETc in "Murcott' mandarin (Table 3). In this regard, the best combination of DI and anti-stressor was DI 85% ETc and Algea 2%, which improved and maintained overall quality and increased water productivity. It is remarkable that plants can adapt to drought by priming variables like ROS, RNS, and chemical agents that may tend to reduce cell water potential and boost water extraction [18,19].

Our results show that applied water and water productivity results agreed with those obtained by El-Tanany et al. [55] and Zaghloul and Moursi [56], who reported that the highest overall mean values of water use efficiency (kg m$^{-3}$) were recorded under irrigation treatments both at 80% and 100% ETc. The effect of Algae 2% extract increasing plant resistance to abiotic stress was also consistent with research by Van Oosten et al. [57].

Similarly, in a study conducted by Mogazy et al. [58], it was shown that plants benefited from a foliar spray containing the microalgae extract when subjected to drought stress.

In general, increasing the SNP concentration increased water productivity regardless of the irrigation treatment (Figure 2). However, trees sprayed with either 100 µmol L$^{-1}$ SNP or 150 µmol L$^{-1}$ SNP under 70% ETc had similar water productivity. An SNP application to plants under well-watered or water-deficient conditions has been shown to increase water productivity by playing a crucial role in several biochemical and developmental processes in crops under common or stressful conditions [59]. Bhuyan et al. [60] showed that an SNP application regulated the water status and enhanced photosynthetic activity, while Majeed et al. [61] reported that an SNP application increased proline content and improved drought tolerance in water-stressed plants.

### 4.2. Fruit Yield and Quality Components

Fruit yield results from non-sprayed control trees under increasing DI regimes were consistent with those published by El-Tanany et al. [55] and Panigrahi et al. [62] on orange trees. Similarly, Hendre et al. [12] showed that water stress to sweet oranges led to a yield decrease. Based on findings by Bousamid et al. [63], with clementine, 100% ETc resulted in an increase in both fruit yield and quality. They also showed that implementing 80% ETc resulted in substantial irrigation water savings. Interestingly, Conesa et al. [64] showed that reduced irrigation to mandarin trees did not affect yield, fruit quality, or storage terms and provided water efficiency in drought conditions. Similarly, 'Star Ruby' grapefruit tree yield and fruit quality were unaffected by the DI technique, which saved 13.2% of water annually [15]. However, in lemon fruit, the DI technique had 31.5% water savings but decreased yield [65].

Current results showed that spraying trees with Algae 2% extract increased 'Murcott' mandarin yield regardless of the DI regime. The microalgae extract has been shown to be a natural bioactive component that stimulates tree growth due to its high levels of auxins and cytokinin [66]. Results were also similar to those by Lall et al. [67], showing the beneficial effects of A. nodosum extract in guava (*Psidium guajava*) fruit production and consistent with those by de Melo et al. [68] and Ebrahimi and Rastegar [69] where the authors highlighted the positive impacts of *Spirulina platensis* on fruit productivity.

Majeed et al. [61] concluded that nitrate assimilation enzyme activities were significantly increased by a SNP spray, which then increased nutrient accumulation and yield under water deficit conditions. These findings highlight the significance of SNP as a stress-signaling molecule that positively regulates defense mechanisms in plants to withstand water-limited conditions. Results from the current study also showed that trees sprayed with 150 µmol L$^{-1}$ SNP under either 85% ETc or 70% ETc had a higher yield when compared to trees under the same DI regime and no anti-stressor spray.

All fruit quality indicators (SSC% and acidity) varied considerably with DI, which agreed with the results of Bousamid et al. [63] and Panigrahi and Srivastava [70]. They also reported Improvements in mandarin fruit volume and juice percentage due to higher irrigation dosages. A post-harvest water loss has been shown to drastically impact fruit durability and marketability [71]. The impact of increasing DI on 'Murcott' fruits can be alleviated by preserving normal cell membrane structure and function by reducing MDA content, which prevents weight loss.

The substantial weight loss of fruits may be attributable to their high MDA content (Table 8), which was indicative of membrane damage due to lipid peroxidation [71] and resulted in fruit water loss and a reduction in fruit weight. This conclusion was reinforced by Wu et al. [72] in table grapes, by Zhao et al. [73] in peaches, by Ge et al. [74] in blueberries, and by Shi et al. [75] in raspberries, and indicated that fruit weight loss percentage was higher in the control plants when compared to fruit weight loss when plants were sprayed with SNP. *Spirulina* also has a wide range of antioxidants, both enzymatic and non-enzymatic [31]. Microalgae antioxidant properties have been shown to slow down the

production of ROS and reduce oxidative stress in mango (*Mangifera indica*) fruit during cold storage, which helped prevent weight loss [68,69].

Ghaffari et al. [76] reported that an increase in osmolytes, particularly proline content, was the most effective strategy to reduce osmotic stress. Additionally, free proline acts as a non-enzymatic radical scavenger to protect the membrane and other macromolecules like enzymes and proteins from damage [77,78]. According to Rezayian et al. [79], NO could increase the osmotic adjustment in plants to enhance their tolerance to drought-induced osmotic stress.

Current results showing that the highest carotenoid content occurred when trees were sprayed with 150 μmol L$^{-1}$ SNP was expected as Li et al. [71] showed that plants sprayed with SNP had significantly less pigment breakdown. In addition, fruits coated with *Spirulina platensis* had significantly less peel color change during storage [69], which suggested that this method may extend the storage life of fruit. Lipid peroxidation in cell membranes was triggered by the generation of free radicals (ROS) in response to various environmental stresses, including drought [76] and low temperature during storage [78].

In response to drought stress, Lau et al. [80] showed that SNPs caused the expression of stress-related genes at the cellular level. The application of SNPs was also shown by Farouk and Al-Huqail [59] to potentially prevent the generation of ROS in response to drought stress and oxidative injury.

Fruit peel MDA content results at harvest and after 15 days of cold storage were comparable to Venkatachalam et al. [81], who showed an increase in MDA concentration during cold storage in both treated and non-treated fruit. They also reported that under chilling stress, non-treated fruit displayed more lipid peroxidation. Treated fruit indicated that the higher the concentration of SNP, the more effective it was against lipid peroxidation. Del Castello et al. [82] also noted that SNP treatments maintained and regulated the levels of ROS by inducing signaling pathways in a series of plant defense-related targets that may detoxify ROS to protect plants. Similarly, Zhao et al. [83] reported that fruits with higher chilling resistance collected lower amounts of MDA. In addition, a nitric oxide treatment was shown to increase fruit's resistance to cold stress via modulating proline metabolism [84,85].

*4.3. Antioxidant Enzyme Activity*

Increased levels of proline were found in plant tissues as a direct consequence of a reaction to various environmental stresses, most notably drought [76], which reinforced the results in the current study. Increased levels of proline were also found in plant tissues as a direct consequence of low temperature during storage [77]. However, in the current study, the proline content did not change after 15 d of cold storage. Wu et al. [72] and Chavoushi [86] concluded that SNP triggered the rise in CAT activity. In addition, Kumar et al. [31] reported that *Spirulina* contained a variety of antioxidants, both enzymatic and non-enzymatic, such as catalase, vitamins, reduced glutathione, total phenol, flavonoid, tannin, carbohydrates, and proteins, and that it may increase CAT and POD activity. Results also agreed with other research that showed enhanced peroxidase activity under water deficit and in response to chilling stress [87].

**5. Conclusions**

This study's findings corroborate previous research suggesting that moderate drought stress can boost fruit quality without impairing the plant's physiological function.

The interactions of 100% ETc and 85% ETc with Algae 2% achieved the highest fruit volume and yield values. But in the water shortage region as the condition of this study, it could be concluded that the interaction of 85% ETc × Algae 2% achieved reasonable values of fruit volume and fruit yield, as well as the highest values of water productivity. This combination had a 51% increase in water productivity compared to non-treated control trees at 100% ETc. 'Murcott' mandarin trees sprayed with higher microalgae or SNP concentrations had fruit with enhanced antioxidant enzyme activity and proline production

when trees were under a water deficit, hence decreasing membrane lipid peroxidation. This resulted in increased cellular membrane function that led to a greater fruit volume and yield. In addition, trees sprayed with the highest SNP concentration had fruit with considerably higher chilling tolerance, which prevented weight loss, when compared to non-treated control trees at the same DI regime. As a result of our research revealed that 'Murcott' mandarin can be successfully irrigated at a lower rate of 85% ETc with Algae at 2% during the entire growing season since the fruits showed no signs of physiological degradation after potential storage life. In addition, it improved product quality regarding soluble solids content. Results suggested that proper irrigation management can result in higher product quality, which would be advantageous if maintained during storage and distributed to the market.

**Author Contributions:** Conceptualization, H.M.E.; methodology, H.M.E., M.M.A.M. and I.F.H.; validation, H.M.E. and M.M.A.M.; formal analysis, H.M.E. and H.M.H.-V.; investigation, H.M.E. and I.F.H.; resources, H.M.E. and H.M.H.-V.; data curation, H.M.E. and I.F.H.; writing—original draft preparation, H.M.E., M.M.A.M., I.F.H. and H.M.H.-V.; writing—review and editing, H.M.H.-V. and H.M.E.; visualization, H.M.E. and M.M.A.M.; supervision, H.M.E. and I.F.H.; funding acquisition, H.M.H.-V. All authors have read and agreed to the published version of the manuscript.

**Funding:** This research received no external funding.

**Institutional Review Board Statement:** Not applicable for studies not involving humans or animals.

**Informed Consent Statement:** Not applicable for studies not involving humans.

**Data Availability Statement:** Please contact Mahmoud Mohamed Abdalla Mahmoud if research data is desired.

**Acknowledgments:** The authors would like to acknowledge Jawahar Jyoti for his statistical advice.

**Conflicts of Interest:** The authors declare no conflict of interest. The funders had no role in the design of this study; in the collection, analyses, or interpretation of data; in the writing of the manuscript; or in the decision to publish the results.

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
