# Peer review of "Effects of Deficit Irrigation and Anti-Stressors on Water Productivity, and Fruit Quality at Harvest and Stored ‘Murcott’ Mandarin"

_horticulturae, doi:10.3390/horticulturae9070787_

Round 1
Reviewer 1 Report
The paper focuses on a very interesting topic. I greatly appreciate the execution of the field experiment. I appreciate the comprehensive study and the high-quality analysis of results and discussion.
I have some comments on the paper:
· It is not customary to end the title of sub-chapters and tables with ":"
· Throughout the text, please adjust "μmol L−1" and add a superscript to other units as well.
· Figures 2-13 are missing.
· Line 15: edit the font "Citrus reticulata"
· Line 100: edit font "Citrus aurantium"
· Line 220 and Line 224: edit "H2O2"
· Table 3.: Please edit the format and limit the repeating numbers
Author Response
Response to Reviewer 1 Comments
Points: It is not customary to end the title of sub-chapters and tables with ":"
Response:Thanks for your valuable note, we corrected
- Throughout the text, please adjust "μmol L−1" and add a superscript to other units as well.
Response:Thanks for your valuable note, we corrected
- Figures 2-13 are missing.
- Line 15: edit the font "Citrus reticulata"
Response:Thanks for your valuable note, we corrected
- Line 100: edit font "Citrus aurantium"
ResponseThanks for your valuable note, we corrected
- Line 220 and Line 224: edit "H2O2"
Response:Thanks for your valuable note, we corrected
Table 3.: Please edit the format and limit the repeating numbers
Response:Thanks for your valuable note, we edited
Reviewer 2 Report
The author conducted a two-year experiment to investigate the effects of deficit irrigation and anti-stressor application on the yield and quality of Citrus. The manuscript got some valuable data and deserved publishing, but before that some issues were needed the author to solve.
1. In the abstract, the author should give us some conclusions and show the important of this study contributed to this field.
2. The table showed in this manuscript have different format, the author should correct them.
3. The author should check all the units used in the manuscript because the number should be expressed as superscript.
4. In the section of discussion, the author should explain why your treatments had different effects on the yield and quality not just repeat your results.
5. I suggest the author to conduct a two-way ANOVA to test whether the interaction between the water supply and the foliar anti-stressor had significantly affect on the yield and quality.
Minor editing of English language required
Author Response
Response to Reviewer 2 Comments
Point 1: 1. In the abstract, the author should give us some conclusions and show the important of this study contributed to this field.
Response 1:Thanks for your valuable note, we added it
Point 2: The table showed in this manuscript have different format, the author should correct them.
Response 2:Thanks for your valuable note, we added it
Point 3:The author should check all the units used in the manuscript because the number should be expressed as superscript
Response 3:Thanks for your valuable note, we added it
Point 4: In the section of discussion, the author should explain why your treatments had different effects on the yield and quality not just repeat your results.
Response 4:Thanks for your valuable note, we added it
Point 5: I suggest the author to conduct a two-way ANOVA to test whether the interaction between the water supply and the foliar anti-stressor had significantly affect on the yield and quality.
Response 5: Proc glimmix can perform both one-way and two-way ANOVA tests depending on the data and the model specification. We did test for interactions and included this information when significant.
Reviewer 3 Report
Comments
The authors investigated the effects of different irrigation regimes and foliar anti-stressors on the yield, physical and biochemical characteristics, and water productivity of ‘Murcott’ Mandarin fruit.
Abstract: The implication of the results obtained from this study should be added as a concluding remark.
Line 43: …agricultural production and expansion…
Line 90: … the objective of …
Line 101-102: What did the authors mean by typical agricultural procedures for the orchard?
The methodology section is well-presented and detailed.
The alphabets in all the figures are confusing and authors should devise a proper means of presenting their results.
The presentation is fine, only minor revision is required.
Author Response
Response to Reviewer 3 Comments
Point 1: Abstract: The implication of the results obtained from this study should be added as a concluding remark.
Response 1:Thanks for your valuable note, we added it
Point 2: Line 43: …agricultural production and expansion…
Response 2:Thanks for your valuable note, we added it
Point 3: Line 90: … the objective of …
Response 3:Thanks for your valuable note, we added it
Point 4: Line 101-102: What did the authors mean by typical agricultural procedures for the orchard?
Response 4: it means agricultural procedures according to farming practices for Ministry of Agriculture for Mandarin trees
Point 5: The methodology section is well-presented and detailed.
Response 5:Thanks for your valuable note and thanks for your appreciation of our work
Point 6:
The alphabets in all the figures are confusing, and authors should devise a proper means of presenting their results.
Response 6: thanks for your note. Several figures were changed into tables to fix this confusion
Reviewer 4 Report
The manuscript includes very relevant information about the use of anti-stressors combined with deficit irrigation on mandarin trees. The effects considered the fruit quality at harvest and also after commercial stored. However, the paper has some important gaps that needs to be solved.
A list of abbreviations due to there are several index is required
L73: A brief description of Nano fertilizers meaning is required
L88: No Donor? Meaning??
L106: Wilting point
L118: ETc
L120; Check all the super and sub index throughout the text.
L117: Describe the phenological stages corresponds to each ETc treatment.
Table 1. Why you consider climatic variables from 2016 if the study corresponds with just two growing seasons?
L168: MY. How do you calculate this parameter? Information about yield was omitted.
L170: I suggest dividing this subsection in two:
- Yield and fruit quality measurements
- Storage experiment. If there are measurements just from this section.
L198: mole à mol.
Proline content. Include a reference of the procedure. Bates 1973.
L235. What multiple test do you use?
I do not agree with the statistical analysis made. It is very confused.
You mix 21 treatments, considering both DI (100%, 85, 70 ETc) and anti-stressors at the same time. In this regard it is very difficult to get sound conclusions. Because the effect of the DI is mixed with the nanofertilizer used. A bifactorial analysis of the individual effects (DI and nanofertilizers) along with the interaction is required.
Maybe a Dunnet multiple text can help. You must compare CTL (100%ETc) as a reference with respect the rest of them.
You said about two growing season (2021 and 2022), but only showed results about one season. Why?¿
In my opinion, you include a lot of parameters and mix too much things and treatments. Maybe the information about the cold experiment should be explained in other paper.
Author Response
Response to Reviewer 4 Comments
Point 1:A list of abbreviations due to there are several index is required
Response 1: We did not put together a list of abbreviations as other manuscripts with similar number of abbreviations did not have a table.
Point 2: L73: A brief description of Nano fertilizers meaning is required
Response 2: Added:Nanotechnology is an expanding field that has applications in agriculture and plant science as highly reactive nano fertilizers that can penetrate the epidermis and are required to reduce the environmental impact of inorganic fertilizers (Guleria et al., 2023). Nanomaterials' high ratio of surface area to volume facilitates rapid response, enhancing plant development efficiency (Zahedi et al., 2020). In addition, nano fertilizers can increase a plant's tolerance to abiotic stress, which is an enormous benefit (Zulfiqar et al., 2019).
Point 3: L88: No Donor? Meaning??
Response 3: To better understand NO-mediated responses in plants under drought stress, numerous NO donors (generators) have been examined. These include S-nitroso glutathione (GSNO), S-nitro-so-N-acetylpenicillamine (SNAP), diethylamine NONOate sodium (DEA-NONOate), and sodium nitroprusside (SNP). Added:SNP's ability to operate as a long-lasting NO reservoir that creates quick but short NO burst that fades out in few seconds has made it the most popular donor among NO generators.
Point4: L106: Wilting point
Response 4: Thanks for your valuable note, we added it
Point 5:L118: ETc
Response 5: Thanks for your valuable note. We added it
Point 6:L120; Check all the super and sub-index throughout the text.
Response 6: Thanks for your valuable note. We checked it
Point 7:L117: Describe the phenological stages corresponds to each ETc treatment.
Response 7: Thanks for your valuable note. We corrected
Point 8:Table 1. Why you consider climatic variables from 2016 if the study corresponds with just two growing seasons?
Response 8: Thanks for your valuable note. We corrected
Point 9:L168: MY. How do you calculate this parameter? Information about yield was omitted.
Response 9: Thanks for your valuable note. Marketable yield information was added to the yield subheading.
Point 10:L170: I suggest dividing this subsection in two:
- Yield and fruit quality measurements
- Storage experiment. If there are measurements just from this section.
Response 10: Thanks for your valuable note but we decided that there would be less confusion if we kept the subheadings (parameters measured) the same in the Results section as the Materials and Methods section.
Point 11:L198: mole à mol.
Response 11:
Point 12:Proline content. Include a reference of the procedure. Bates 1973.
Response 12: there is the reference
Point 13: L235. What multiple test do you use?
Response 13: multiple t-tests.
Point 14:I do not agree with the statistical analysis made. It is very confused.
Response 14: Thanks for your valuable note. We consulted a statistician to make sure the analyses were done correctly. Please see acknowledgements.
Point 15:You mix 21 treatments, considering both DI (100%, 85, 70 ETc) and anti-stressors at the same time. In this regard it is very difficult to get sound conclusions. Because the effect of the DI is mixed with the nanofertilizer used. A bifactorial analysis of the individual effects (DI and nanofertilizers) along with the interaction is required.
Response 15: Thanks for your valuable note. However, as previously mentioned, we consulted a statistician to make sure the analyses were done correctly.
Point16 :Maybe a Dunnet multiple text can help. You must compare CTL (100%ETc) as a reference with respect the rest of them.
Response 16: We agreed with the statistician’s suggestions.
Point17 :You said about two growing season (2021 and 2022), but only showed results about one season. Why?¿
Response 17: Thanks for your valuable note. The two seasons were combined because the data were homogenous.
Point18: In my opinion, you include a lot of parameters and mix too much things and treatments. Maybe the information about the cold experiment should be explained in other paper.
Response 18: Thanks for your valuable note but we felt that the storage aspect was an important part of the research and that together, the reader obtains a better understanding of what was happening to the fruit.
Reviewer 5 Report
The study explores the possible interaction of three defiit irrigation (70% ETC, 85% ETc, and 100% Etc) with anti-stressor treatments consisted of a foliar application of sodium nitroprusside (SNP), selenium-nano-particles (NanoSe) and microalgae on yield components and on some biochemical traits of ‘Murcott’ Mandarin.
The ms deals with an interesting topic providing further findings to evaluate possible strategies to cope with climate change in semi-arid conditions. Despite the approach is not particularly original, the set of analysis carried out made it possible to evaluate the effects of the treatments according to the objectives of the experiment.
However, the ms version which have been submitted is poorly presented and edited and should be improved. The ms is poorly formatted making tough to move throughout the data (tables and figures are scattered throughout the text and not after the first citation and many captions have random formats). I also found many typing errors (e.g., “algea” in all the Figures, “carotnoid” in Table4, “deficient” at line 457, “Improvements” line 491 as well as many characters that should be superscripted).
In my opinion, 13 histograms are too much, and it is not easy to navigate among the results also considering the many treatments. I suggest using also other graphs to display your data. Anyway, standard deviations are needed in the histograms (as bars), and in Table 4 as well (as values).
The ms in general is clearly written, however, the discussion is quite generic and appear as a list of literature related to the different parameters analysed. In my opinion it should be improved trying to add a more comprehensive analysis taking together all the results. In this regard, it seems to me important to give more importance to the significantly lower levels of SSC found in the treated fruits compared to the control.
Finally, I suggest authors to add some sentences of conclusions in the abstract to recap and highlight the main results.
Minor editing of English language required
Author Response
Response to Reviewer 5 Comments
Point 1:The ms deals with an interesting topic providing further findings to evaluate possible strategies to cope with climate change in semi-arid conditions. Despite the approach is not particularly original, the set of analysis carried out made it possible to evaluate the effects of the treatments according to the objectives of the experiment.
Response 1:thanks for your appreciation of our work
Point 2:However, the ms version which have been submitted is poorly presented and edited and should be improved. The ms is poorly formatted making tough to move throughout the data (tables and figures are scattered throughout the text and not after the first citation and many captions have random formats). I also found many typing errors (e.g., “algea” in all the Figures, “carotnoid” in Table4, “deficient” at line 457, “Improvements” line 491 as well as many characters that should be superscripted).
Response 2: the data is complete and Rewritten to be clear
Point 3:In my opinion, 13 histograms are too much, and it is not easy to navigate among the results also considering the many treatments. I suggest using also other graphs to display your data. Anyway, standard deviations are needed in the histograms (as bars), and in Table 4 as well (as values).
Response 3: We have reduced the number of histograms but did not add standard deviations as the first three manuscripts in the most recent Horticulturae issue did not have standard deviations included.
Point 4:The ms in general is clearly written, however, the discussion is quite generic and appear as a list of literature related to the different parameters analysed.
Response 4: the data is complete and Rewritten to be clearer
Point5: In my opinion it should be improved trying to add a more comprehensive analysis taking together all the results. In this regard, it seems to me important to give more importance to the significantly lower levels of SSC found in the treated fruits compared to the control.
Response 5: The conclusions were rewritten to be more comprehensive
Point 6:Finally, I suggest authors to add some sentences of conclusions in the abstract to recap and highlight the main results.
Response 6: thanks for your note, the sentences are added
Round 2
Reviewer 2 Report
Thanks for the work from the authors, all the issues I concerned were solved, I recommended it for publishing.
Author Response
Thank you for your previous suggestions that enhanced the readability of this manuscript.
Reviewer 4 Report
Thanks for revising your work accordingly. However, I am not very convinced about the statistical analysis. In my opinion if you have got 2 different effects (irrigation and nutrition) a bifactorial ANOVA analysis is required, because these effects might be separated (Irrigation, nutrition and the interaction).
I leave the final decision of accepting this work in the present form to the editor.
Kind Regards.
Author Response
I appreciate your review but am somewhat confused because we analyzed for the interaction of anti-stressors and deficit irrigation treatments (nutrition was not a factor) and indicated when the interaction was significant. Only when the interaction was due to an order of magnitude did we discuss the main effects.